# Enhanced Anticancer Activity of *Hymenocardia acida* Stem Bark Extract Loaded into PLGA Nanoparticles

**DOI:** 10.3390/ph15050535

**Published:** 2022-04-26

**Authors:** Oluwasegun Adedokun, Epole N. Ntungwe, Cláudia Viegas, Bunyamin Adesina Ayinde, Luciano Barboni, Filippo Maggi, Lucilia Saraiva, Patrícia Rijo, Pedro Fonte

**Affiliations:** 1Department of Pharmacognosy, Igbinedion University, Benin 23401, Nigeria; adedokun.oluwasegun@iuokada.edu.ng; 2Research Center for Biosciences & Health Technologies (CBIOS), Universidade Lusófona de Humanidades e Tecnologias, 1749-024 Lisboa, Portugal; epole.ntungwe@ulusofona.pt; 3Department of Biomedical Sciences, Faculty of Pharmacy, University of Alcalá de Henares, 28805 Alcalá de Henares, Spain; 4Department of Chemistry and Pharmacy, Faculty of Sciences and Technology, University of Algarve, Gambelas Campus, 8005-139 Faro, Portugal; viegas.claudiasofia@gmail.com; 5Center for Marine Sciences (CCMAR), University of Algarve, 8005-139 Faro, Portugal; 6Faculty of Medicine and Biomedical Sciences (FMCB), University of Algarve, 8005-139 Faro, Portugal; 7Department of Pharmacognosy, University of Benin, Benin 23401, Nigeria; baayinde@uniben.edu; 8School of Science and Technology, Chemistry Division, University of Camerino, 62032 Camerino, Italy; luciano.barboni@unicam.it; 9Chemistry Interdisciplinary Project (ChIP), School of Pharmacy, University of Camerino, 62032 Camerino, Italy; filippo.maggi@unicam.it; 10LAQV/REQUIMTE, Laboratόrio de Microbiologia, Departamento de Ciências Biolόgicas, Faculdade de Farmácia, Universidade do Porto, 4050-313 Porto, Portugal; lucilia.saraiva@ff.up.pt; 11Instituto de Investigação do Medicamento (iMed.ULisboa), Faculdade de Farmácia, Universidade de Lisboa, 1649-003 Lisboa, Portugal; 12iBB-Institute for Bioengineering and Biosciences, Department of Bioengineering, Instituto, Superior Técnico, Universidade de Lisboa, 1049-001 Lisboa, Portugal; 13Associate Laboratory i4HB–Institute for Health and Bioeconomy at Instituto Superior Técnico, Universidade de Lisboa, Av. Rovisco Pais, 1049-001 Lisboa, Portugal

**Keywords:** anticancer activity, cytotoxicity, *Hymenocardia acida*, nanoencapsulation, nanoparticle, plant extract, PLGA

## Abstract

*Hymenocardia acida (H. acida)* is an African well-known shrub recognized for numerous medicinal properties, including its cancer management potential. The advent of nanotechnology in delivering bioactive medicinal plant extract with poor solubility has improved the drug delivery system, for a better therapeutic value of several drugs from natural origins. This study aimed to evaluate the anticancer properties of *H. acida* using human lung (H460), breast (MCF-7), and colon (HCT 116) cancer cell lines as well as the production, characterization, and cytotoxicity study of *H. acida* loaded into PLGA nanoparticles. Benchtop models of *Saccharomyces cerevisiae* and *Raniceps ranninus* were used for preliminary toxicity evaluation. Notable cytotoxic activity in benchtop models and human cancer cell lines was observed for *H. acida* crude extract. The PLGA nanoparticles loading *H. acida* had a size of about 200 nm and an association efficiency of above 60%, making them suitable to be delivered by different routes. The outcomes from this research showed that *H. acida* has anticancer activity as claimed from an ethnomedical point of view; however, a loss in activity was noted upon encapsulation, due to the sustained release of the drug.

## 1. Introduction

Cancer is generally referred to as a lethal disease characterized by the uncontrolled growth and replication of cancer cells. It can occur in most organs of the multicellular organism and is reported as one of the major public health challenges. Additionally, it is the principal cause of morbidity worldwide among all age groups [1,2,3,4]. Cancer is the second leading cause of death in developing countries and the leading cause of death in developed countries [5]. Population growth, aging as well as the adoption of lifestyles associated with smoking, drinking, lack of physical exercise, and consumption of chemically contaminated foods have caused an increase in cancer incidence in developing nations [2,6,7]. Statistical reports show that by the year 2000, 10 million new cases of cancer had emerged with an increase of 25% every decade. Mortality rates associated with cancer might increase from 6 million to 16 million between the years 2000 and 2050, with 17 million and 7 million novel cases from developing and developed countries, respectively [8,9,10].

The management of cancer has been challenging despite the numerous methods of modern treatments available. These include radiotherapy, surgery, immunotherapy, and chemotherapy which can be used alone or in combination. Localized cancers are usually treated by surgery and radiation while cancer cells that have metastasized to other parts of the body are treated using chemotherapy, such as alkylating agents, antibiotics, hormones, and antimetabolites [8,10,11].

Despite the cytotoxic attributes of chemotherapeutic agents, they have significant limitations. For example, they display numerous side effects by affecting proliferating normal cells localized in the hair, bone marrow, and gastrointestinal tract. Other limitations include low absorption rate, development of secondary malignancy, high cost of drug/treatment, insolubility, instability, and tumor drug resistance.

All these limitations impose the search for natural drugs with improved efficacy, selectivity, reduced toxicity, and low secondary effects inherent in cancer management [12].

Plants are natural sources of drugs and have been used as medicines for at least 60,000 years. They can produce secondary metabolites with a wide range of pharmacological properties, including anticancer activity. They can be used as crude and/or their derived natural products or compounds and have been useful in cancer treatment, research, and development [13,14,15]. *Hymemocardia acida* Tul (Hymenocardiaceae) is a dioecious and deciduous shrub, mostly found in the Savannah region of the Southwestern part of Nigeria, normally 6–10 m in height. It is characterized by contorted and stunted growth, and it is widely known and used in African trado-medicine. It is called “heart-fruit” in English, “enanche” in Idoma, “ikalaga” in Igbo, “ii-kwarto” in Tiv, “emela” in Etulo, “Uchuo” in Igede, “jan yaro” in Hausa, “yawa satoje” in Fulani, and “Orunpa” in Yoruba [16,17,18]. Ethnomedicinal information suggests that the plant is used traditionally to treat hemorrhoids, chest pain, eye infection, migraine, skin diseases, and several infections, and as a poultice to treat abscesses and tumors [16,17,19]. Phytochemical studies indicate that these therapeutic applications result from their varied composition of secondary metabolites such as alkaloids, terpenoids, glycosides, flavonoids, saponins, and tannins [17].

Although experimental findings have shown that many natural products have a strong therapeutic value, their poor solubility and bioavailability (at the target organ) have been a challenge over time. Another problem associated with the use of conventional plant extract-based formulations is the presence of toxicity to other organs and tissues. To overcome this, some scientists have used the “green chemistry” approach to nanoparticle production that includes clean, non-toxic, and environmentally friendly methods. NP synthesized via green synthetic routes are highly water soluble, biocompatible, and less toxic [20]. Other strategies using hybrid systems combining nanoparticles and ionic liquids may be also used to improve the delivery of poorly soluble drugs [21,22]. 

Nanotechnology by the nanoencapsulation of natural products in a polymer to improve drug delivery to cancer targets has gained considerable interest over the past decades [17,23,24]. Thus, polymer-based drug delivery systems allow the control of drug release, enhance effective drug solubility, minimize drug degradation, contribute to reduced drug toxicity, and facilitate control of drug uptake, which significantly contributes to the therapeutic efficiency of a drug. Poly (lactic-co-glycolic acid) (PLGA) based nanoencapsulation has been shown to possess numerous advantages over other conventional delivery devices based on high biocompatibility, biodegradability, drug protection from degradation, sustained and controlled drug release, linkage of other molecules with PLGA for better interaction with biological materials, and the possibility to target specific organs or cells. Furthermore, PLGA will be degraded into nontoxic substances and the breakdown products are lactic acid and glycolic acid, which are hydrophilic, diffusible, and rapidly metabolized in the human body [25,26,27]. PLGA has also shown good results in improving the bioavailability of drugs delivered by the oral route, a non-invasive route that may be promising in cancer treatment [28], hence, the choice of PLGA base encapsulation in this research.

This research, therefore, aims to evaluate the cytotoxic activity of crude *H. acida* methanol stem extract using both benchtop assays as well as human cancer cell lines (breast, colon, and lung cancer cell lines). Additionally, nanoencapsulation of the extract was carried out using the sulforhodamine B (SRB) assay and reviewing comparative studies on the nanoencapsulated extract and crude extract on breast, colon, and lung cancer cell lines. This method allows the determination of the cell density, based on the measurement of cellular protein content and the cytotoxicity screening of compounds with adherent effect to 96-well format [29]. 

## 2. Results and Discussion

Surgery and or radiotherapy have been great tools for the management of diverse forms of cancer if diagnosed early. Studies have shown that half of all cancer patients use some form of integrative therapy when cancer cells are not responding to medical procedures to reduce pain as well as improve the overall wellbeing of the patients. 

The use of medicinal plants for the treatment of diverse diseases, including cancer, cannot be overemphasized as they have served as the source for compounds of therapeutic importance [30]. Medicinal plants are a source for lead compounds and highly bioactive drugs useful in the management of diseases associated with man and animals [30,31]. Recent studies have shown that 55% of chemotherapeutics are directly or indirectly from natural products [32]. Side effects associated with present cancer therapeutics and increasing cancer cases have prompted the search for novel anticancer agents of plant extract or isolated compounds of natural origin, which needs to be studied using both in vitro and in vivo cytotoxicity models [30,33].

### 2.1. In Vitro Cytotoxicity Assay of H. acida Using R. ranninus

In this study, tadpoles (*R. ranninus*) were used in the in vitro cytotoxicity assay of *H. acida* crude extract due to their accessibility, mostly in the rainy season, allowing the simulation of a complete multicellular organism. The *H. acida* extract was exposed to *R*. *ranninus* for 24 h at a concentration range of 20–400 μg/mL. The cytotoxic potential against this model was verified by a reduction in the movement of the tadpoles and confirmed with consequent cessation of movement. The results reveal a significant difference in cytotoxicity activity (*p* ≤ 0.05) in all concentrations of *H. acida* examined relative to 5% DMSO (negative control), which has no cytotoxic effect on *R*. *ranninus*. Moreover, at a concentration of 20 μg/mL, the *H. acida* extract showed 89.52 ± 1.52% bioactivity while concentrations at 40–400 μg/mL indicated 100.00 ± 0.00% cytotoxic potential against *R*. *ranninus* (Figure 1). 

### 2.2. In Vitro Cytotoxicity Activity of Crude H. Acida Using S. cerevisiae

The yeast *S. cerevisiae* is one of the widely used eukaryotic models. Its rapid growth and ease of manipulation to evaluate multiple biological effects induced by the drugs under consideration make it a suitable model for cytotoxicity study [34]. In vitro preliminary cytotoxicity study on the crude stem bark extract of *H. acida* was performed against *S. cerevisiae* using nystatin as positive control and 2% DMSO in YPD as blank. The growth rate of *S. cerevisiae* in the blank was considered to be 100%, that is, a zero percentage of inhibition, so its absorbance was maximum at 300 min (0.328). The cells were exposed to *H. acida* extract and nystatin at different concentrations (7.81 to 500 µg/mL) and the absorbance of the specific cell growth rates was measured from 0 to 300 min. 

According to the results (Table 1), *H. acida* extract showed a concentration-dependent effect. The extract significantly inhibited *S. cerevisiae* growth in all the concentrations tested over time when compared with Nystatin and negative control (DMSO). Overall, the percentage growth inhibition ranges from 71.70 to 100%. At a concentration of 500 µg/mL, 100% of inhibition was observed similar to Nystatin. 

Considering that the higher the percentage growth of inhibition, the higher the general toxicity of the extract, we can conclude that *H. acida* extract is toxic against S. *cerevisiae* and *R. ranninus*. The correlation between the results of these two organisms validates their use for preliminary toxicity studies. These results can be supported by the composition of *H. acida* in cyclopeptide alkaloids, namely in hymenocardine, found by Tuenter et al. (2016). In their studies, this compound present in the root bark of *H. acida* showed cytotoxicity activity against MRC-5 cells (human lung fibroblasts) with an IC_50_ value of 51.1 ± 17.2 μM [35]. Moreover, an in vivo study carried out by Sowemimo et al. (2007) showed that *H. acida* steam bark extract is toxic to brine shrimps and caused chromosomal damage in rat lymphocytes, and consequently that it is mutagenic and has cytotoxic activity [31].

### 2.3. Phytochemical Study of H. acida

To understand the chemical composition of the bioactive *H. acida* crude extract, and to identify the main compound responsible for the tested bioactivity, this extract was subjected to several chromatographic techniques. 3β- lup-20(29)-en-3ol (Lupeol) was isolated from this extract (as a colorless crystal, mp 212–214 °C) and its structure was confirmed through a comparison of its spectroscopic data (Appendix A) to those described in the literature (Figure 2) [36,37,38].

### 2.4. H. acida-Loaded PLGA Nanoparticles Production and Characterization

PLGA nanoparticles are used to improve the pharmacokinetics, stability, and delivery of the extract [33]. Therefore, to enhance the drug delivery and therapeutic potentials of *H. acida* crude extract, *H. acida* nanoparticles (HA-Np) and blank nanoparticles (unloaded Np, negative control) were produced and characterized [39].

The mean hydrodynamic particle size of both HA-Np and unloaded Np were 210 ± 3 nm and 193 ± 2 nm, respectively, which showed good method robustness and ability to obtain a nanoparticle size suitable for different delivery routes [40,41,42]. The nanoparticles were further observed by scanning electron microscopy (SEM) to confirm the nanoparticles size and evaluate its morphology (Figure 3). The nanoparticles presented a spherical shape and smooth surface characteristic of PLGA nanoparticles [43]. No relevant differences were observed between unloaded and HA-Np, demonstrating the robustness of the production method.

The Pdl of the nanoparticles was also determined. Small values of PdI (near to zero) were desirable because this indicates a uniform size distribution and a monodisperse nanoparticle formulation [44]. In the case of HA-Np, a PdI of 0.231 ± 0.050 was obtained to 0.100 ± 0.010 observed in unloaded Np, which implies more heterogeneity between the HA-Np particles and a polydisperse formulation as shown in Table 2. Similar results for nanoparticle size distribution and PdI were obtained by our group in the encapsulation of other drugs [42,45]. Although, this is an expected known result for loaded Np because the particles have to contain the volume of the extract of *H. acida* [40]. 

The diffusion constant describes the quantity of a substance that is diffusing from one region to another through a unit cross-section per unit time when the volume–concentration gradient is constant. A higher diffusion constant of 2.34 × 10^8^ ± 0.07 was observed in *H. acida* nanoparticles (HA-Np) relative to unloaded Np 2.55 × 10^8^ ± 0.09, which implies a faster rate of diffusion due to the small particle size of HA-Np. Although the diffusion constant is a physical constant that depends on molecular size, temperature (high surface area to volume ratio), pressure, and other properties of the diffusing substance, a reduced diffusion constant of HA-Np will enhance rapid contact of nanoparticle to the targeted receptor for improved drug delivery. Additionally, both unloaded Np as well as HA-Np nanoparticles obtained were homogenous in aspect and form a homogenous colloidal formulation. Moreover, 61.71 ± 2.17% association efficiency was observed, which is a very good achievement. A similar refractive index of 1.33 ± 0.01 was observed in both blank-Np as well as HA-Np, which implies that light waves will pass through both particles in a vacuum by 1.3328 times slower, which also showed a good nanoparticle formulation as shown in Table 2.

To confirm that the extract is incorporated in the polymeric matrix of the PLGA nanoparticles and to assess drug–polymer interactions upon encapsulation, FTIR analysis of *H. acida* extract, unloaded Np, and HA-Np was carried out (Figure 4). The FTIR analysis is a powerful non-invasive technique to assess the structure of NP and its content [46]. Their data also confirms that the extract is incorporated in the polymeric matrix of the PLGA nanoparticles because the transmittance band in the range 3100–3600 cm^−1^ present in the *H. acida* extract is reflected slightly in the HA-Np spectrum. Another characteristic band of the extract is found at 1600 cm^−1^ in the HA-Np spectrum. On the other hand, in the HA-Np spectrum, the bands related to the nanoparticles at the 1000–1600 cm^−1^ zone are attenuated. All these bands confirm that *H. acida* extract is incorporated in the polymeric matrix of the PLGA nanoparticles. It is also possible to check the spectra of both unloaded Np as well as HA-Np, the intense band relative to the carbonyl groups present in the two monomers of PLGA (C = O stretching vibrations) around 1750 cm^−1^, the band relative to ester bond (C-O-C stretching vibrations) around 1186 cm^−1^ and the band relative to C–H stretches around 2285–3010 cm^−1^ which does not appear in the *H. acida* extract spectrum [25,40].

### 2.5. Cytotoxic Effect of H. acida and PLGA Nanoparticles on Human Cancer Cell Lines

PLGA is an FDA-approved polymer known for its biomedical applications in drug delivery due to its versatility, biodegradability, and biocompatibility. It is used extensively to prepare nanoparticles to deliver a wide range of therapeutic agents, including active pharmacological molecules, peptides, and nucleic acids [47]. 

The PLGA nanoparticles were produced to protect the *H. acida* extracts and to allow controlled release of the extracts into the target cells. Therefore, it is important that the stability and also cytotoxic effect of the *H. acida* extracts are maintained and that the release of the contents of the PLGA nanoparticles occurs promptly. The in vitro cytotoxicity of the *H. acida* extract and HA-Np nanoparticles was assessed using Sulfordiamine (SRB) assay. The results for the cytotoxic effect of *H. acida* crude extract and HA-Np on colon colorectal carcinoma (H460), human breast adenocarcinoma (MCF-7), and lung cancer carcinoma (HCT116) using this assay are shown in Table 3. 

*H. acida* crude extract had an IC_50_ (µg/mL) of 20.80 ± 6.10 in the human lung (H460) cancer cell line, 38.70 ± 0.80 in MCF-7, and 42.90 ± 0.20, in colon (HCT116) cancer cell lines. The results obtained for cancer cell lines subjected to *H. acida* extract reveal a good cytotoxic effect of this extract, specifically in the H460 human lung cancer cell line. These results are in comparison with those obtained by Calhelha et al. [48]. The cytotoxic effect (in GI_50_ values, µg/mL) of Portuguese propolis samples (collected in Aljezur) against the lung (NCI-H460), breast (MCF7), and colon (HCT15) cancer cell lines (37 ± 1, 47 ± 2, and 50 ± 11) once again affirm the potential that *H. acida* extract is cytotoxic. Those results from *H. acida* were also comparable with the study proceeded by Sharma et al. [49], which reveals similar results from the anticancer activity of essential oil from *Cymbopogon flexuosus* in lung cancer cell lines (IC_50_ values varied from 49.7 to 79.0 µg/mL for each line) and in colon cancer cell lines (IC_50_ values varied from 4.2–60.2 µg/mL for each line). Thus, although the mechanisms by which *H. acida* has cytotoxic effects are unknown, it appears that its extracts have an impact on cell viability, and thus in cancer treatment. This impact of the extract on cell viability could be partly explained due to the presence of the isolated lupeol. Lupeol is shown to have cytotoxicity in different cancer cell lines. The anti-leukemic activity of this compound was tested against the K562 cells and it was shown to decrease cell viability [50]. Similar results were observed in other studies against different cancer cell lines where lupeol was cytotoxic against MCF-7, Caco-2, SW620, KATO-III, HCT-116 cell lines [51,52,53,54]. Lupeol can thus contribute to the cytotoxicity of *H. acida* extracts.

However, an IC_50_ > 50 for HA-Nps was observed in all human cancer cell lines, which means a poor activity of *H. acida* loaded in PLGA nanoparticles. One hypothesis for this loss of inactivity might be a result of the delayed release of the drugs (*H. acida*) from the PLGA nanoparticles. To overcome this loss of activity, a lower PLGA concentration could be used since this will result in a thinner cover of the Nps, and the production of highly porous nanoparticles making it easier to release the content [26]. Another hypothesis for this loss of cytotoxic effect of *H. acida* loaded in PLGA nanoparticles could be due to some loss of stability of the extracts during the encapsulation process, and consequently loss of efficacy.

Contrary to our results for the activity of HA-Np, recently, Adlravan et al., in their study on the potential cytotoxic activity of *Nasturtium officinale* extract non-nanoencapsulated (free NOE) and PLGA/PEG nanoencapsulated (NOE-loaded) in human lung carcinoma A549 cells, found that NOE-loaded showed better cytotoxic effects than free NOE. This work reinforces the idea that the nanoencapsulation of the extracts improves the anticancer effects of the therapies, as well as allows a sustained and controlled release of NOE constituents from nanoparticles and increases intracellular concentrations. On the other hand, free NOE easily diffuses through the lipid bilayers, being more rapidly eliminated, leading to lower cytotoxicity on target cells [55].

Based on the above results, additional studies should be performed on the in vitro release study for HA-Np to understand if the extract is difficult to release from the HA-Np. Other concentrations of PLGA or combinations of polymers should also be studied since encapsulation of extracts into nanoparticles is known to be a promising strategy to enhance therapeutic efficiency, and consequently overcome these challenges. 

## 3. Materials and Methods

### 3.1. Materials and Cell Lines 

(3-(4,5-dimethyl-2-thiazolyl)-2,5-diphenyl-2H-tetrazolium bromide) (MTT), rhodamine 123 (Rho123), 5-fluorouracil, Pluronic F-68, and dimethylsulfoxide (DMSO) were from Sigma–Aldrich Chemie GmbH (Paris, France). PLGA Resomer^®^ RG 503 H (was obtained from Evonik Industries, Essen, Germany. Phosphate buffer saline (PBS) was from Merck (Darmstadt, Germany). Fetal bovine serum (FBS) was from Gibco, Alfagene, Carcavelos, Portugal. RPMI-1640 medium (Roswell Park Memorial Institut), DMEM (Dulbecco’s Modified Eagle Medium), penicillin–streptomycin solution, antibiotic–antimycotic solution, L-glutamine, and trypsin/EDTA were from PAA (Vienna, Austria). *Saccharomyces cerevisiae* (ATCC 2601) cell culture, yeast extract peptone (YPD), HCT-116 (lung), MCF-7 (breast), and H460 (colon) human cancer cell lines were from the National Cancer Institute (Frederick, MD, USA). All other chemicals used in this study were of analytical grade and were purchased locally.

#### Cell Culture Maintenance

The cells were grown and maintained in an appropriate medium, pH 7.4, supplemented with 10% fetal calf serum, glutamine (2 mM), penicillin (100 units/mL), and streptomycin (100 μg/mL). The cell cultures were grown in a carbon dioxide incubator (Heraeus, GmbH, Germany) at 37 °C with 90% humidity and 5% CO_2_ [56,57]. 

All cancer cells were cultured in RPMI-1640 medium with ultraglutamine (Lonza, VWR, Carnaxide, Portugal), and supplemented with 10% FBS. Cells were maintained at 37 °C in a humidified atmosphere of 5% CO_2_.

### 3.2. Botanical Authentication and Extraction

*H. acida* stem barks were collected from the Iwo community in Osun State, Nigeria. Botanical identification and authentication were carried out at the herbarium section of the University of Benin by Prof. MacDonald Idu (Professor of Phytomedicine and Taxonomy). The voucher specimen (UBH-R633) was deposited at the herbarium unit. The plant was grounded to a coarse powder using a laboratory milling machine. The extraction was carried out in methanol using 1.2 kg of the plant powder and the plant extract was obtained using a Soxhlet apparatus. The crude extract obtained was concentrated using Heidolph Rotavapor (LABORATA 4000) with a speed set at 120 rpm and a reduced temperature of 40 °C. The concentrated extract was removed from the round bottom flask with methanol and poured into weighed beakers [58].

### 3.3. In Vitro Cytotoxicity Assay Using R. ranninus (Tadpoles)

A preliminary cytotoxicity study was carried out on crude stem extract of *H. acida* using *R. ranninus*. The organisms were collected from pounds at Olomo beach, Uhonmora village, Edo State. Ten *R. ranninus* of similar sizes were placed into different beakers containing 30 mL of the freshwater from the habitat of tadpoles. The volume was completed up to 49 mL with distilled water and the extract was added to a total volume of 50 mL. The extract was tested at 20, 40, 100, 200, and 400 µg/mL dissolved in 5% DMSO. The experimental procedure was performed in triplicate and a control assay was performed using 50 mL containing 1 mL of 5% DMSO in distilled water [23,56,59]. The mortality rates of the tadpoles were observed for a maximum of 24 h.

### 3.4. Isolation and Structural Characterization of Lupeol

About 51.90 g of aqueous fraction of *H. acida* was subjected to vacuum liquid chromatography (VLC) using dichloromethane, ethylactetae, and methanol in increasing order of polarity. This yielded four (A to D) VLC fractions, based on similarities in their analytical TLC profile, A (1; 1.13 g), B (2–3; 1.89 g), C (4–6; 4.88 g), and D (7–8; 41.15 g). B was further fractionated by normal phase open column chromatography using Silica gel G (kieselgel 70–230 mesh size) and dichloromethane, ethylactetae, and methanol as eluent with increasing polarity. Detection was carried out using non-destructive (visible light and UV light (254 and 365 nm)) followed by spraying with concentrated sulphuric acid and heating at 110 °C). This resulted in seven fractions (BF12–8). Fraction BF3 obtained from column chromatography was subjected to a series of purification using preparative-TLC and this resulted in a colorless crystal, lupeol (12.4 mg).

The 1D and 2D NMR analysis of the compound were carried out using a Bruker Fourier spectrometer (600 MHz). The compound was dissolved in deuterated chloroform. ^1^H and ^13^C chemical shifts are expressed in part per million (ppm) while coupling constant (*J*) as Hertz (Hz) (Appendix A).

### 3.5. In Vitro Cytotoxicity Assay Using Saccharomyces Cerevisiae

Further preliminary cytotoxicity study was carried out on the crude stem extract of *H. acida* using *Saccharomyces cerevisiae* (*S. cerevisiae*). Approximately 1.0 × 10^7^ cells per mL of *S. cerevisiae* cell cultures were obtained by inoculating *S. cerevisiae* grown on YPD medium (yeast extract 1%, peptone 0.5%, and glucose 2%) containing 1.5% agar into 20 mL of YPD and placed into an incubator 30 °C without agitation for 16–20 h. About 0.5 × 10^6^ cells were transferred into 4 mL disposable cuvettes containing YPD medium and aliquots of stock solution of plant extract to obtain concentrations of 7.81 µg/mL, 15.6 µg/mL, 31.2 µg/mL, 62.5 µg/mL, 125 µg/mL, 250 µg/mL, and 500 µg/mL to a total volume of 2.2 mL. Nystatin, a known antifungal was used as the positive control while YPD medium and 5% DMSO were used as the negative controls. The cuvettes were incubated in a Heidolph Incubator 1000 with a shaker at 30 °C and 230 rpm agitation to ensure homogeneous suspensions for 5 h. Initial absorbance was measured at the start time (0 min) and every 60 min. The assay was performed in triplicates for each concentration. The reproducibility of the results was analyzed by repeating the assay on three different days. The absorbance at 525 nm of each sample (cell cultures) over the time (0–300 min) was measured. Growth curves were obtained from the number of cells per mL of YPD medium over time; the percentage growth inhibition rate of *S. cerevisiae* in the presence of *H. acida* stem extract and nystatin was determined. Statistical analysis was performed using one-way analysis of variance (ANOVA) and the Krustal–Wallis test (non-parametric) for comparison between groups. The values are presented as mean ± SEM; significant difference at *p* < 0.05 was considered. [57].

### 3.6. Production of H. acida Loaded PLGA Nanoparticles

*H. acida* loaded PLGA nanoparticles (HA-Np) were produced by solvent-evaporation o/w single emulsion technique [60]. Crude extract containing 20 mg of *H. acida* was added to 5 mL acetone: methanol (8:2), along with 50 mg PLGA resulting in *H. acida* organic solution. Then, this organic phase solution was added in a dropwise manner into a 10 mL aqueous solution containing the stabilizer Pluronic F-68 1% (*w*/*v*). This mixture was sonicated for 30 s at 70% of amplitude in a Q125 Sonicator (QSonica Sonicators, Newtown, CT, USA). The formed emulsion was then subjected to evaporation under reduced pressure for organic solvent removal. The formulations were washed three times and resuspended in ultrapure water. Then, the samples were freeze-dried for further use. Blank nanoparticles (unloaded Np) were also produced following the same procedure.

### 3.7. H. acida Loaded PLGA Nanoparticles Characterization

The freeze-dried samples were reconstituted with ultrapure water at the desired concentration and were lightly shaken in a vortex for 2 min for complete homogenization. The mean hydrodynamic particle size, polydispersity index (PdI), diffusion constant (D) and refractive index, and viscosity (cP) were evaluated by dynamic light scattering (DLS) using a Malvern Zetasizer Nano ZS ZS (Malvern Instruments, UK). Each sample of unloaded Np and HA-Np formulation was analyzed in triplicate at 25 °C.

The drug loading into PLGA nanoparticles was quantified by evaluating the association efficiency percentage (%AE) by an indirect method, where the amount of *H. acida* encapsulated into PLGA nanoparticles was calculated by the difference between the total amount of *H. acida* extract used in the nanoparticle formulation and considering the free *H. acida* amount in the supernatant after centrifugation of HA-Np formulation in HERMLE Z323K ultracentrifuge at 15,000× *g* during 20 min at 4 °C. The quantification of free *H. acida* in the supernatant was performed by Folin–Ciocalteu’s method by using a UV-Visible spectrophotometer. 

The %AE of HA-Np was determined by the following Equation (1):(1)$$\%\mathrm{AE}=\frac{Total\;amount\;of\;H.acida-Free\;H.acida\;in\;supernatant}{Total\;amount\;of\;H.acida}\times 100$$

The morphology of the PLGA nanoparticles was evaluated by SEM using a FEI Quanta 400 FEG SEM (FEI, Hillsboro, OR, USA). In a prior observation, the nanoparticles were placed on metal stubs, and vacuum-coated with a layer of Gold/Palladium for 60 s with a current of 15 mA.

### 3.8. Fourier Transform Infrared Spectroscopy Spectroscopy 

The *H. acida* extract, HA-Np, and unloaded Np were evaluated by ATR-FTIR. All spectra were collected from 64 scans, in the 4000–500 cm^−1^ range at 4 cm^−1^ resolutions, on an ABB MB3000 FTIR (Zurich, Switzerland). All spectra were area-normalized for comparison using the Origin 8 software (OriginLab Corporation, Northampton, MA, USA). 

### 3.9. In Vitro Cytotoxicity Assay against Human Cancer Cell Lines

The crude extract and nanoencapsulated *H. acida* extract were subjected to in vitro cytotoxicity assay using human cancer cell lines involving semiautomatic procedure using sulforhodamine-B (SRB) assay, as described earlier [29,30,61,62]. They were tested in different cancer cell lines: colon colorectal carcinoma (HCT116), human breast adenocarcinoma (MCF-7), and lung cancer carcinoma (H460). The procedure involves growing human cancer cell lines in tissue culture flasks at a temperature of 37 °C, 5% CO_2_ as well as 90% relative humidity in a complete growth medium. Flasks with a subconfluent stage of growth were selected and cells were harvested by treatment with trypsin-EDTA. 

Cells were plated in 96-well plates at a density of 10,000 cells/100 μL cells/well and incubated for 24 h. *H. acida* and encapsulated samples were added to the 96-well plates. The extracts were tested at 10, 30, and 100 μg/mL, and prepared in DMSO (the final DMSO concentrations were between 0.001% (lowest) and 0.5% (highest)). The effect of the samples was analyzed following 48 h incubation, using the sulforhodamine B (SRB) assay. Briefly, following fixation with 10% trichloroacetic acid from Scharlau (Sigma–Aldrich, Sintra, Portugal), plates were stained with 0.4% SRB from Sigma–Aldrich (Sintra, Portugal) and washed with 1% acetic acid. The bound dye was then solubilized with 10 mM Tris Base and the absorbance was measured at 540 nm in a microplate reader (Biotek Instruments Inc., Synergy, MX, USA). The concentration of *H acida* and nanoencapsulated *H. acida* extract that causes a 50% reduction in the net protein increase in cells (IC_50_) was determined. Data are mean ± SEM of 4–5 independent experiments [62].

### 3.10. Statistical Analysis

All data collected from the entire study were analyzed using *Microsoft Excel* and GraphPad Prism 7 (developed by Dr. Harvey Motulsky, San Diego, USA). Relevant tables, charts, and descriptive statistics were used to present the pertinent points of the study. Data were expressed as the mean ± SEM. The data were subjected to statistical analysis using one-way analysis of variance (ANOVA) and complemented with the Krustal–Wallis test (non-parametric).

## 4. Conclusions

*H. acida* possesses significant toxicity in *S. cerevisiae* and *R. ranninus models.* It had *the* highest cytotoxicity (IC_50_ of 20.80 ± 6.10 µg/mL) against the lung cancer cell lines. The solubility of this extract was successfully improved through nanoencapsulation. However, a loss in cytotoxicity was observed with IC_50_ = >50 for all the human cancer cell lines tested. This may be due to the sustained delay in the release of the extract from the nanoencapsulation. The present results show that *H. acida* can be a promising source for possible anticancer compounds. Further research is ongoing to identify more bioactive principles using bio-guided isolation procedures, identify the mechanism of action and structure–activity relationship in the bioactive principle(s), and improve the methods for encapsulation and controlled delivery.

## Figures and Tables

**Figure 1 pharmaceuticals-15-00535-f001:**
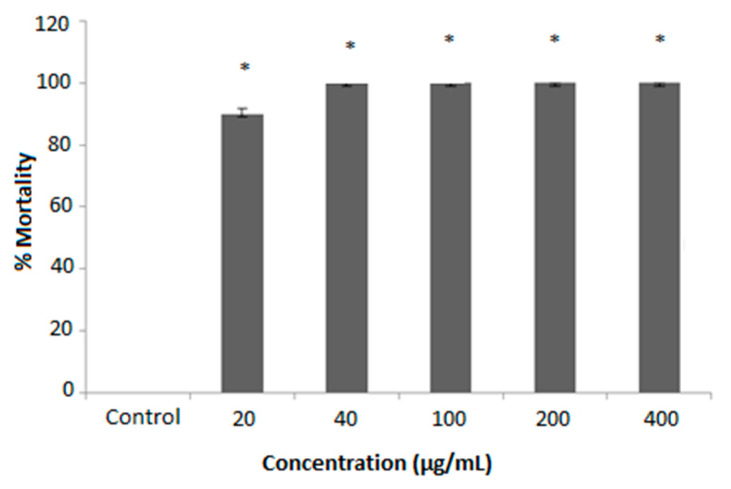
Effect of crude *Hymenocardia acida* (*H. acida*) extract on % *Raniceps ranninus* (*R. ranninus*) mortality at concentrations ranging from 20–400 µg/mL–an index of cytotoxic effect. A 5% DMSO solution was used as negative control. Each bar represents the mean ± SEM of 10 (ten) independent experiments (*n* = 10). Samples with superscript * indicate a significant difference at *p* < 0.05 relative to the negative control.

**Figure 2 pharmaceuticals-15-00535-f002:**
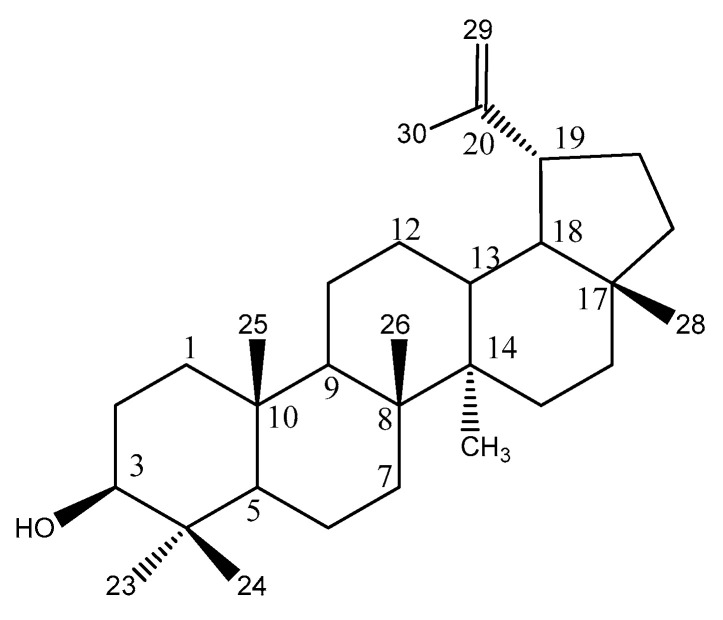
Lupeol isolated from *H. acida*.

**Figure 3 pharmaceuticals-15-00535-f003:**
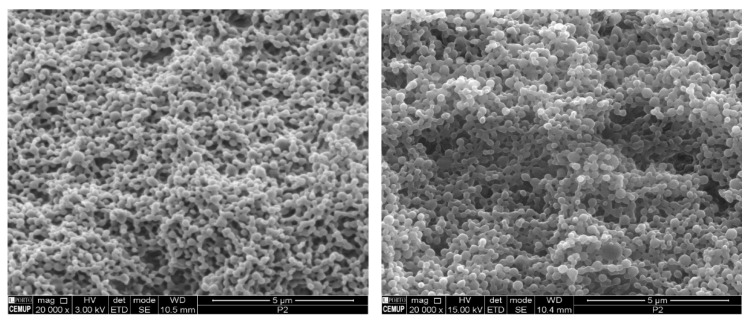
SEM microphotographs of unloaded PLGA Np (**left**) and *H. acida*-loaded PLGA nanoparticles (**right**).

**Figure 4 pharmaceuticals-15-00535-f004:**
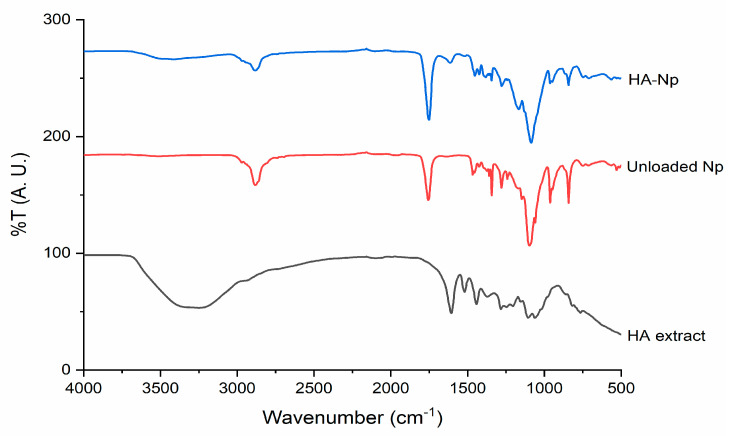
ATR-FTIR spectra of crude extract of *H. acida* (HA extract), blank nanoparticles (unloaded Np), and *H. acida* nanoparticles (HA-Np).

**Table 1 pharmaceuticals-15-00535-t001:** General toxicity effect of crude extract of *H. acida* on the percentage of growth inhibition of *S. cerevisiae*.

Concentration (µg/mL)	% Growth Inhibition		
	DMSO ^a^	Nystatin ^b^	*H. acida* Crude
7.81	12.67 ± 1.21	97.25 ± 1.02 *	90.30 ± 0.99 *
15.6	16.80 ± 1.08	98.21 ± 0.98 *	71.70 ± 1.12
31.2	17.60 ± 0.01	98.78 ± 2.17 *	95.70 ± 1.10 *
62.5	30.73 ± 1.12	99.35 ± 2.92 *	95.40 ± 2.08 *
125	31.20 ± 1.03	99.59 ± 1.87 *	96.56 ± 1.98 *
250	33.84 ± 1.03	99.71 ± 1.34 *	96.98 ± 2.11 *
500	33.91 ± 1.10	100.00 ± 0.00 *	100.00 ± 0.00 *

The values above are presented by mean ± SEM of three replicates (*n* = 3). Values with superscript * indicate a significant difference at *p* < 0.05 when compared to the corresponding percentage inhibition of solvent (DMSO ^a^) for each concentration using one-way analysis of variance (ANOVA) and complemented with the Krustal–Wallis test (non-parametric), ^b^ = positive control, and ^a^ = negative control.

**Table 2 pharmaceuticals-15-00535-t002:** Physical–chemical properties and characterization of blank nanoparticles (unloaded Np) and *H. acida* nanoparticles (HA-Np) (*n* = 3, mean ± SEM). Results are significantly different (*p* < 0.05).

Parameter	Unloaded Np	HA-Np
Particle size (nm)	210 ± 3	193 ± 2
Polydispersity índex (PdI)	0.100 ± 0.010	0.231 ± 0.050
%AE	Not Applicable	61.71 ± 2.17%
Homogeneity	Homogenous	Homogenous
Colour	Whitish	Milky
Diffusion constant (D) (cm^2^/sec)	2.34 × 10^8^ ± 0.07	2.55 × 10^8^ ± 0.09
Refractive Index	1.33 ± 0.01	1.33 ± 0.11
Viscosity (cP)	0.890 ± 0.110	0.888 ± 0.170

**Table 3 pharmaceuticals-15-00535-t003:** Cytotoxic effect (IC_50_ (µg/mL)) of *H. acida* nanoparticles in H460, MCF-7, and HCT116 cell lines of *H. acida* and *H. acida* nanoparticles using sulforhodamine B assay after 48 h of treatment. Data are presented by mean ± SEM (*n* = 4).

	Cancer Cell Lines (IC_50_ (µg/mL))		
	H460	MCF-7	HCT116
*H. acida* crude	20.80 ± 6.10	38.70 ± 0.80	42.90 ± 0.20
HA-Np	>50	>50	>50
Doxorubicin	0.29 ± 2.32	0.08 ± 4.10	0.05 ± 3.24

## Data Availability

Data is contained within the article and Appendix A.

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
