# Peer review of "Enhanced Anticancer Activity of Hymenocardia acida Stem Bark Extract Loaded into PLGA Nanoparticles"

_pharmaceuticals, 2022, doi:10.3390/ph15050535_

Round 1

Reviewer 1 Report

The MS is well written and described an important medicine for cancer therapy.  I have mentioned several comments directly on the MS which will be helpful for authors to make MS more clear. However, my major concern is about replication and repetition of the experiments. I never found the term replication mentioned in the MS. In addition, to draw the reliable conclusion and remove biological errors, each experiment must be repeated. Did authors repeat experiments, if yes then mention it and also how did you handle the data from repeated experiments for analysis. This should be clarified by editor with author before publication.   

Author Response

Pharmaceuticals (ISSN 1424-8247)

Manuscript ID

pharmaceuticals-1658898

Type: Article

Title: Enhanced Anticancer Activity of Hymenocardia acida Stem Bark Extract Loaded into PLGA Nanoparticles

Authors: Oluwasegun Adedokun , Epole Ntungwe , Claudia Viegas , Bunyamin Adesina Ayinde , Luciano Barboni , Filippo Maggi , Lucília Saraiva , Patrícia RIJO * , Pedro Fonte

Section: Natural Products

Special Issue: Plant and Marine-Derived Natural Product Research in Drug Discovery: Strengths and Perspective

Reviewer 1 Comments:

The MS is well written and described an important medicine for cancer therapy.  I have mentioned several comments directly on the MS which will be helpful for authors to make MS more clear. However, my major concern is about replication and repetition of the experiments. I never found the term replication mentioned in the MS. In addition, to draw the reliable conclusion and remove biological errors, each experiment must be repeated. Did authors repeat experiments, if yes then mention it and how did you handle the data from repeated experiments for analysis. This should be clarified by editor with author before publication.  

We thank the reviewer for the comments. All experiments in this manuscript were done in triplicates and for the cytotoxicity using the sulforhodamine-B assay, 4–5 independent experiments were done. This is heighted in yellow in each of the biological activity assays in the material and method section. The statistical analysis of the data is mentioned in each assay and also described in the section 3.10. Statistical analysis.

Reviewer 2 Report

The title of the work is Enhanced Anticancer Activity of Hymenocardia acida Stem 2 Bark Extract Loaded into PLGA Nanoparticles. The authors prepare PLGA Nanoparticles and uploaded extract on the nanoparticles. This work is interesting but There are many things to be included for example the authors should characterized the nanoparticles after ;loading extract by using XRD, SEM, TEM, AFM etc. techniques. Besides, a comparison of the extract activity should be made with nanoparticle loaded extract. Is there any effect of nanoparticles on the normal cells?

The English is poor and needs improvement.

Literature is not enough and following refs may give the strength to this manuscript.

RSC Adv., 9, 15357-15369 (2019)

My decision is a major revision.

Author Response

Pharmaceuticals (ISSN 1424-8247)

Manuscript ID

pharmaceuticals-1658898

Type: Article

Title: Enhanced Anticancer Activity of Hymenocardia acida Stem Bark Extract Loaded into PLGA Nanoparticles

Authors: Oluwasegun Adedokun , Epole Ntungwe , Claudia Viegas , Bunyamin Adesina Ayinde , Luciano Barboni , Filippo Maggi , Lucília Saraiva , Patrícia RIJO * , Pedro Fonte

Section: Natural Products

Special Issue: Plant and Marine-Derived Natural Product Research in Drug Discovery: Strengths and Perspective

Reviewer 2 Comments:

1) The title of the work is Enhanced Anticancer Activity of Hymenocardia acida Stem 2 Bark Extract Loaded into PLGA Nanoparticles. The authors prepare PLGA Nanoparticles and uploaded extract on the nanoparticles. This work is interesting but There are many things to be included for example the authors should characterized the nanoparticles after; loading extract by using XRD, SEM, TEM, AFM etc. techniques. Besides, a comparison of the extract activity should be made with nanoparticle loaded extract. Is there any effect of nanoparticles on the normal cells?

We thank the reviewer for the suggestion. Accordingly, we performed SEM analysis to characterize the nanoparticles morphology and the discussion of the results was added to the manuscript. The cytotoxicity of the H. acida extract and HA-Np nanoparticles was assessed using SRB assay. The results showed a loss in cytotoxicity in the nanoparticles loaded extract, which may be due to the sustained release of the drug. The effect of these nanoparticles on the normal cells will be assessed and the results included in another manuscript.  This manuscript is a preliminary study and the work is ongoing.

2) The English is poor and needs improvement.

We thank the reviewer for the comment. The English were corrected by a native English speaker

3) Literature is not enough and following refs may give the strength to this manuscript.

RSC Adv., 9, 15357-15369 (2019)

We thank the reviewer for the comment. The literature was reviewed and more references were added.

My decision is a major revision.

Reviewer 3 Report

It is an interesting work. However, there are still some problems that should be addressed carefully before the final acceptance. Major revision is needed.

Special comments for the revision:

  1. It is necessary for the authors to indicate clearly the novelty and significance of this work.
  2. In the section of 2.1, the authors could provide corresponding microscopy images of R. ranninus by adding H. acida extract with various concentrations.
  3. In the section of 2.4, it is necessary for the authors to add more characterization data on PLGA and H. acida-loaded PLGA nanoparticles. For instance, the authors could use TEM or SEM to measure the size of PLGA.
  4. In the section of 2.5, it is necessary for the authors to provide additional fluorescence microscopy images of cells to identify the cytotoxic effect of both unloaded and loaded PLGA nanoparticles.
  5. More discussion on the advantages of the acida extract for the modification of PLGA nanoparticles for biomedical application should be added.

Author Response

Pharmaceuticals (ISSN 1424-8247)

Manuscript ID

pharmaceuticals-1658898

Type: Article

Title: Enhanced Anticancer Activity of Hymenocardia acida Stem Bark Extract Loaded into PLGA Nanoparticles

Authors: Oluwasegun Adedokun , Epole Ntungwe , Claudia Viegas , Bunyamin Adesina Ayinde , Luciano Barboni , Filippo Maggi , Lucília Saraiva , Patrícia RIJO * , Pedro Fonte

Section: Natural Products

Special Issue: Plant and Marine-Derived Natural Product Research in Drug Discovery: Strengths and Perspective

Reviewer 3 Comments:

It is an interesting work. However, there are still some problems that should be addressed carefully before the final acceptance. Major revision is needed.

Special comments for the revision:

1) It is necessary for the authors to indicate clearly the novelty and significance of this work.

We thank the reviewer for the suggestion. That was indicated in the abstract and introduction sections of this work. “H. acida is used traditionally for the treatment of various ailments including cancer. This study aimed to evaluate the anticancer properties of H. acida using human lung (H460), breast (MCF-7), and colon (HCT 116) cancer cell lines as well as production, characterization, and cytotoxicity study of H. acida loaded into PLGA nanoparticles”.

2) In the section of 2.1, the authors could provide corresponding microscopy images of R. ranninus by adding H. acida extract with various concentrations.

We thank the reviewer for the comment we add the microscopy images.

3) In the section of 2.4, it is necessary for the authors to add more characterization data on PLGA and H. acida-loaded PLGA nanoparticles. For instance, the authors could use TEM or SEM to measure the size of PLGA.

We thank the reviewer for the comment. We evaluated the size of PLGA nanoparticles using DLS, which is the standard technique for this purpose, since SEM or TEM do not allow to precisely evaluate the nanoparticles size. Still, we performed SEM analysis to observe and characterize the nanoparticles morphology, so that data was added to the manuscript.

4) In the section of 2.5, it is necessary for the authors to provide additional fluorescence microscopy images of cells to identify the cytotoxic effect of both unloaded and loaded PLGA nanoparticles.

We thank the reviewer for the comment. Since the present study is preliminary, the data regarding fluorescence microscopy will be regarded in a future manuscript

5) More discussion on the advantages of the acida extract for the modification of PLGA nanoparticles for biomedical application should be added.

We thank the reviewer for the comment. More discussion on the advantages of the H. acida PLGA nanoparticles for biomedical application was added.

Round 2

Reviewer 1 Report

Authors addressed my all comments raised in old version.

Author Response

We include all the corrections in the new version, and highlight all the corrections and changes.

Reviewer 3 Report

In this revised version, the authors addressed most of the questions that indicated by the referees. Now the manuscript is recommended for publication in present form.

Author Response

We improved the manuscript as suggestted by the reviewer.